# Grape Pomace Extracts as Fermentation Medium for the Production of Potential Biopreservation Compounds

**DOI:** 10.3390/foods8020051

**Published:** 2019-02-02

**Authors:** Maxwell Mewa-Ngongang, Heinrich W. du Plessis, Seteno K. O. Ntwampe, Boredi S. Chidi, Ucrecia F. Hutchinson, Lukhanyo Mekuto, Neil P. Jolly

**Affiliations:** 1Post-Harvest and Agro-Processing Technologies, ARC Infruitec-Nietvoorbij (The Fruit, Vine and Wine Institute of the Agricultural Research Council), Private Bag X5026, Stellenbosch 7599, South Africa; DPlessisHe@arc.agric.za (H.W.d.P.); chidib@arc.agric.za (B.S.C.); Hutchinsonu@arc.agric.za (U.F.H.); JollyN@arc.agric.za (N.P.J.); 2Bioresource Engineering Research Group (BioERG), Department of Biotechnology, Cape Peninsula University of Technology, P.O. Box 652, Cape Town 8000, South Africa; NtwampeS@cput.ac.za (S.K.O.N.); lukhayo.mekuto@gmail.com (L.M.)

**Keywords:** growth inhibition activity, production kinetics, optimization, potential biopreservation compounds, *Candida pyralidae*, *Pichia kluyveri*

## Abstract

Microbial spoilage causes food losses in the food industry and as such, the use of synthetic chemical preservatives is still required. The current study proposes the use of agro-waste, i.e., grape pomace extracts (GPE), as production medium for biopreservation compounds. Production kinetics, subsequent to optimization using response surface methodology (RSM) for biopreservation compounds production was studied for three yeasts using GPE broth as a fermentation medium. The results showed that the highest volumetric zone of inhibition (VZI) was 1.24 L contaminated solidified media (CSM) per mL biopreservation compounds used (BCU) when *Candida pyralidae* Y1117 was inoculated in a pH 3-diluted GPE broth (150 g L^−1^) incubated at 25 °C for 24 h. Similar conditions were applied for *Pichia kluyveri* Y1125 and *P. kluyveri* Y1164, albeit under slightly elongated fermentation periods (up to 28 h), prior to the attainment of a maximum VZI of only 0.72 and 0.76 L CSM mL^−1^ ACU, respectively. The potential biopreservation compounds produced were identified to be isoamyl acetate, isoamyl alcohol, 2-phenyl ethylacetate and 2-phenyl ethanol.

## 1. Introduction

The loss of fruit and beverages due to microbial spoilage impacts negatively on the economy of the producing countries [1,2]. In developing countries, such losses are severe due to poor conservation and transportation facilities available to producers [3]. Some of the common microorganisms that are associated with fruit and beverage spoilage are *Botrytis*, *Colletotricum*, *Rhizopus*, *Dekkera*, *Zygosaccharomyces*, *Pichia* and *Hanseniaspora* species [4,5,6]. The current methods for the preservation of beverages and fruits are mainly based on synthetic chemicals. However, due to the serious health concerns associated with the use of synthetic chemicals [7], there is an urgent need for their replacement with less harmful preservatives, which are a healthier alternative for humans and cost effective. Recently, yeasts were identified as potential producers of biopreservation compounds and potential biocontrol agents against several spoilage organisms [2,8,9,10,11]. These investigations lacked an industrial engineering approach for sustainable production of these biopreservation compounds, and used expensive refined media that is not cost-effective [12]. However, yeasts also have the ability to produce useful metabolites while growing in inexpensive media [13,14,15].

Grape pomace extract (GPE) is an inexpensive potential raw material that could be used for the production of value-added products like biopreservation compounds. South Africa has a thriving wine industry [16] and large quantities of grape pomace are generated, essentially as a waste product. The chemical composition of GPE could make a suitable alternative medium for the growth of various microorganisms, since it contains fermentable reducible sugars [17] such as glucose, fructose and yeast-assimilable nitrogen (YAN) that can fulfil the nutritional requirements for microbial growth of producers of biopreservatives.

During microbial growth, the relationship between growth rates and the production kinetics of extracellular metabolites, is key in assessing the general physiological requirements of yeasts, including substrate utilization and extracellular metabolites formation rates. The process for the production of biopreservation compounds needs to be optimized for higher product yield and minimal cost of production. Optimization approaches such as response surface methodology (RSM) and Box-Behnken have been used previously [18,19,20]. However, based on the number of experimental runs required when developing suitable response surface models, the central composite design is the most preferred design with regard to response surface methodology [20,21,22].

As supported by the literature, the nature of the growth inhibition by yeasts have been attributed to its higher growth rate, which makes them competitive for nutrient and space [14]. Another attribute is the production of extracellular compounds such as killer toxins and volatile organic compounds (VOCs) [9,10,14]. Since the current literature has demonstrated the potential of yeasts as producers of VOCs with growth inhibition properties mostly at the screening level without much production using renewable bioresources, this research forms part of a larger project investigating the use of wild yeasts as producers of biopreservation compounds. This study’s aim was to understand the kinetics of biopreservation compound production by *C. pyralidae* and *P. kluyveri* and to optimize the production process, using grape pomace extracts, a cheap, readily available raw material, as a fermentation medium.

## 2. Materials and Methods

### 2.1. Yeast Culture Conditions and Inoculum Preparation

Yeasts previously identified and screened for growth inhibition activity against common spoilage yeasts (Table 1) were obtained from the culture collection of the Agricultural Research Council (ARC Infruitec-Nietvoorbij, Stellenbosch, South Africa). A volume of 50 mL of pre-inoculum of each of the biopreservation compounds producing yeasts (Table 1) was prepared separately in Yeast Peptone Dextrose (YPD) broth (Biolab, Merck, South Africa) and the pH adjusted to 5 using 0.1 M HCl. The inoculated 100 mL flasks containing 50 mL broth were incubated at 25 °C and agitated at 150× rpm for 48 h. The spoilage yeasts were also grown in YPD broth as described above. Samples (2 mL) were centrifuged at 5000× rpm for 5 min and the pellets recovered were washed with sterile distilled water. The pellets were re-suspended in 1 mL of sterile distilled water and the cell concentrations determined using a Haemocytometer at 400× magnification.

### 2.2. Grape Pomace Extracts Medium Preparation

Wet grape pomace from Chenin Blanc berries was obtained from the ARC Infruitec-Nietvoobij research cellar and pressed at 200 kPa in order to extract the remaining juice from the wet pomace. The resulting juice was racked and the GPE broth obtained and frozen in plastic buckets at −10 °C. Prior to use, the grape pomace extract was thawed and diluted with water to sugar concentrations of 100, 150 and 200 g L^−1^. After the dilutions, the yeast assimilable nitrogen (YAN) was measured using an enzyme robot (Arena 20XT; Thermo Fisher Scientific, Vantaa, Finland) and found to be sufficient to support the growth of yeasts. The different dilutions were adjusted to pH 5, using 0.1 M NaOH and immediately autoclaved for 30 min at 120 °C.

### 2.3. Concept of Volumetric Zone of Inhibition and Calculation

The concept of volumetric zone of inhibition was developed using Equation (1):$$A=\pi r^{2}\;(r\;=\;D/2);\;V=A\cdot H$$ where *H*, *A*, *V* represent the thickness, area and the volume covered by the grape pomace extract agar, respectively. Prior to determining this volume, $A=\pi r^{2}$ (*r* = D/2) was used to calculate the area covered by the grape pomace extract agar (GPE) agar, where *r* and *D* represent the radius, and the diameter of the petri dish, respectively.

The data obtained from the measured zone of inhibition, the thickness of the GPE agar and the diameter of the pierced wells on the GPE agar were used in Equation (1). The diameter zone of inhibition (*D* = *D_o_* − *d*) measured around the well (Figure 1) is a resultant of the volume of 20 µL of the biopreservation compounds used (BCU) which was introduced in the pierced well.

This volume (20 µL) of the inoculated, solidified GPE agar was the concept basis of the volumetric zone of inhibition (VZI). Since 1 cm^3^ = 1 mL = 10^−3^ L, the calculated volume (*V* = *A*·*H*) units expressed in liters (L) was used for consistency. In this study, the VZI interpretation was based on the fact that 20 µL (0.02 mL) of biopreservation compounds sample was sufficient to inhibit the growth of the spoilage organism inoculated at 1 × 10^6^ CFU mL^−1^ in a defined volume (L) of the GPE agar plate. The term inhibitory activity (L mL^−1^) was then adopted to describe the growth inhibitory effects which are reported as the volume of contaminated (inoculated) solidified media (L CSM) per mL of biopreservation compounds used (mL BCU). The units were presented as L CSM mL^−1^ BCU.

### 2.4. Preliminary Assessments for Biopreservation Compounds Production, Chemical Analysis and Growth Inhibition Assay

The preliminary assessment for production of biopreservation compounds was done using Chenin Blanc GPE. A concentration of 10^6^ cells mL^−1^ of *C. pyralidae* Y1117, *P. kluyveri* Y1125 and *P. kluyveri* Y1164 were inoculated separately (in triplicate) in 250 mL Erlenmeyer flasks, containing 150 mL of the autoclaved GPE broth at a concentration of 100, 150 and 200 g L^−1^. The flasks were incubated for 32 h at 25 °C in a shaking incubator (LM-53OR, RKC^®^ Instrument INC, Ohta-ku Tokyo, Japan) set at 150× rpm. Samples were withdrawn every 4 h, centrifuged at 5000× rpm for 5 min and the supernatants were filtered using 0.22 µm sterile nylon membrane filters. Filtered samples (1 mL) were analyzed for total sugar and YAN while the remaining aliquots were used for the growth inhibition assay.

The growth inhibition assay was adapted from [10] with some modifications. The GPE agar used for growth inhibition assay was prepared by supplementing the GPE broth (150 g·L^−1^) with 2% agar bacteriological (Biolab, Merck, South Africa).

A well with a 5-mm diameter was drilled on GPE agar plates using an agar driller. Prior to drilling, the GPE agar plates were seeded with 10^6^ cells mL^−1^ of either *Z. bailli*, *D. bruxellensis* or *D. anomala* [10]. In the agar wells, a volume of 20 µL of the crude biopreservation compounds was spotted, subsequent to incubation at 22 °C until clear zones of inhibition were observed around the 5 mm wells. The plates were then assessed for biopreservation activity, which was quantified as described by Mewa-Ngongang et al. [23].

### 2.5. Kinetic Studies for Production of Potential Biopreservation Compounds from GPE

The production conditions under which the kinetic study was carried out is described in Section 2.3. Key parameters, such as the rates of substrate utilization, biomass formation, specific growth, biopreservation compounds formation, including that based on cell concentration and substrate consumption and biomass yield, were used. The samples that were withdrawn every 4 h under the fermentation conditions described in Section 2.3, were analyzed for sugar, cell concentration and product formation, subsequent to fitting the data in the selected existing models. The level of product formation was assessed by the size of the volumetric zone of inhibition of each sample tested. The total sugar utilization models were used, as described in Appendix A, in order to assess the efficiency of GPE broth as a fermentation medium. The microbial growth dependency of biopreservation compounds production was determined using the modified Malthus equation [24], with the specific growth rate of individual yeasts being determined under the defined experimental conditions (Section 2.3).

### 2.6. Response Surface Methodology (RSM) for the Optimization of Biopreservation Compounds Production using GPE Broth as Fermentation Medium

A central composite design (CCD) approach was used and a total of 30 experimental runs for each yeast was generated, using Design-Expert^®^ software version 10.0.0 (Stat-Ease Inc., Minneapolis, MN, USA), to assess the effect of independent variables (fermentation time, pH, temperature and total sugar concentration) on the production of potential biopreservation compounds. The independent variable interactions were determined by fitting the experimental data to a second order polynomial model (Equation (1) in Appendix A). Each experiment had three replicates and the mean value of each run was used for data fitting while accounting for variations in the experimental data. Appendix A contains the process variables used and their ranges.

The statistical analysis was used to determine the significance of the models generated for each yeast strain. It was carried out by means of analysis of variance (ANOVA) incorporated in the Design-Expert^®^ software version 10.0.0 used. Furthermore, numerical optimization software incorporated in Expert design version 10.0.0 was also used to identify the interactions of independent variables that yielded the highest concentration of potential biopreservation compounds (Appendix A).

### 2.7. Identification and Quantification of VOCs Produced by *C. pyralidae* Y1117, *P. kluyveri* Y1125 and *P. kluyveri* Y1164

Aliquots (10 mL) of fresh juice was placed in a 20 mL headspace vial to which NaCl (30% *m*/*v*) was added to facilitate evolution of volatiles into headspace and inhibit enzymatic degradation. Vials were spiked with 100 µL of anisole d8 and 3-octanol as internal standards. Solid-Phase Micro Extraction (SPME) vials were equilibrated for 5 min in the CTC autosampler incubator (50 °C) at 250× rpm. Subsequently, a 50/30 divinylbenzene/-carboxen/-polydimethylsiloxane (DVB/CAR/PDMS) coated fiber was exposed to the sample headspace for 10 min at 50 °C. After the VOCs’ adsorption onto the fiber extraction, desorption of the VOCs from the fiber coating was carried out in the injection port of the gas chromatography-mass spectrometry (GC–MS) for 10 min. The fiber was inserted in a fiber conditioning station for 10 min between samples for cleaning to prevent cross- and carry-over contamination. Chromatographic separation of the VOCs was performed in a Thermo TRACE 1310 gas chromatograph coupled with a Thermo TSQ 8000 mass spectrometer detector. The GC–MS system was equipped with a polar DB-FFAP column (Model number: J&W 122-3263), which is a nitroterephthalic-acid-modified polyethylene glycol (PEG) column of high polarity for the analysis of VOCs, with a nominal length of 60 m; 250-μm internal diameter; and 0.5-μm film thickness. Analyses were conducted using helium as a carrier gas at a flow of 2.9 mL min^−1^. The injector temperature was maintained at 250 °C. The oven program was as follows: 350 °C for 17 min; and subjected to a final temperature of 240 °C at an increased rate of 12 °C min^−1^ and held for 6 min. The MS was operated in a full scan mode. Both the ion source transfer line temperatures were maintained at 250 °C. Compounds were tentatively identified by comparison with a mass spectral libraries (NIST, version 2.0), subsequent to quantification using the calculated relative abundances.

## 3. Results and Discussion

### 3.1. Growth Inhibition Assay on Beverage Spoilage Yeasts

*Candida pyralidae* Y1117 showed growth inhibition activity against all three beverage spoilage organisms (*D. bruxellensis*, *D. anomala* and *Z. baillii*), while the two *P. kluyveri* strains only showed inhibition activity against *D. bruxellensis* and *D. anomala* (Figure 2). Mehlomakulu et al. [10] reported growth inhibition activity of *C. pyralidae* against *D. bruxellensis*, while this study showed the inhibition activity of *C. pyralidae* against *Z. baillii* and *D. anomala* as well (Figure 2). According to the literature reviewed, some growth inhibition activities are associated with killer toxins produced by yeasts, including *Candida pyralidae* [10]. However, this study paid a special attention to the VOCs as well as the quantification aspects of the overall growth inhibition activity obtained using cheaply available raw material as fermentation medium.

### 3.2. Fermentation Kinetics of Potential Biopreservation Compounds Produced in GPE Broth

The highest growth inhibition activity for *C. pyralidae* (0.797 L CSM mL^−1^ BCU) was obtained at 150 g L^−1^ after 24 h of fermentation, 20 to 28 h for *P. kluyveri* Y1164 (0.412 L CSM mL^−1^ BCU) and 28 h for *P. kluyveri* Y1125 (0.373 L CSM mL^−1^ BCU) (Figure 3). Based on the current observations, the potential of diluted residual GPE as a cheap raw material for the production of possible biopreservation compounds from yeasts was tentatively shown.

*C. pyralidae* Y1117 had the lowest substrate utilization rate (0.333 g L^−1^ h^−1^) compared to *P. kluyveri* Y1125 and *P. kluyveri* Y1164 (Table 2). In general, the substrate utilization rate and biomass yield were inversely proportional to the rate of biopreservation compounds formation. In addition, minor differences in specific growth rate for *C. pyralidae* Y1117 (0.196 h^−1^), *P. kluyveri* Y1125 (0.202 h^−1^) and *P. kluyveri* Y1164 (0.190 h^−1^) were observed. The findings in Table 2 are also indicative of a direct relationship between the formation rate of biopreservation compounds and biomass yield. Overall, the substrate utilization model showed that cellular growth and production of biopreservation compounds were directly linked to sugar utilization rates for all the yeasts. Similarly, Mewa-Ngongang et al. [22] previously established a direct relationship between biopreservation compounds formation and substrate utilization rate. However, in this study, *C. pyralidae* yielded more biopreservation compounds, while utilizing less substrate, a trait that can be exploited at industrial scale.

### 3.3. Response Surface, Model Validation and Optimum Conditions for the Production of Biopreservation Compounds

The interactive effect of the four independent parameters (fermentation time, pH, temperature and total sugar concentration) on production of biopreservation compounds is represented in a 3D plot (Figure 4). An interdependence of fermentation time, temperature and sugar concentration on the production of biopreservation compounds was observed. It was noted that the optimal conditions for production of biopreservation compounds were both pH and temperature dependent. Therefore, optimal production was only obtained at pH lower than 5 and temperature between 15 and 25 °C for all three yeasts strains studied (Figure 4a-1,b-1,c-1). These conditions typify mild and low cost conditions, as these bioreactor conditions would be easily maintained.

Previously, Ngongang et al. [12] showed that temperature and pH had a significant effect on cellular growth subsequent to the production of biopreservation compounds. The optimal production time for the biopreservation compounds ranged between 16 and 24 h for *C. pyralidae* Y1117, and between 16 to 28 h for *P. kluyveri* Y1125 and Y1164 (Figure 4a-2,b-2,c-2). Variations in sugar concentration of the production medium had an insignificant effect on biopreservation compound production. However, a combinatorial effect of fermentation period and sugar concentrations of the GPE broth on production of biopreservation compounds was noted when both conditions (viz. ‘prolonged fermentations in higher sugar must’ and ‘shorter fermentations in lower sugar must’) were responsible for higher growth inhibition activity (Figure 4a-3,b-3,c-3). It was then economically realistic to consider the use of lower sugar containing broth (diluted GPE broth to 150 g L^−1^), which still showed the highest growth inhibition activity rapidly (±24 h) for *C. pyralidae* Y1117 (Figure 4a-3). Contrarily, *P. kluyveri* Y1125 (Figure 4b-3) and Y1164 (Figure 4c-3) seemed to have higher physiological requirements that were characterized by longer fermentation periods and higher sugar requirements. The significance of fermentation parameters such as pH, temperature and carbon source for production of biopreservation compounds was also highlighted for *Lactobacillus* and *Saccharomyces* species [25,26,27]. In this study, the determination of the optimum conditions for production of biopreservation compounds in a GPE medium was done based on the desirability function. Insignificant variations in biopreservation compound production were noted when both ‘low sugar concentration-short fermentation time’ and ‘high sugar concentration-prolonged fermentation time’ settings were compared for all strains (Appendix A).

Three different quadratic models (*Y*_1_, *Y*_2_ and *Y*_3_) were generated to explain the production of biopreservation compounds in GPE medium. Based on the analysis of variance for each model, it was found that all models were significant with a probability value of <0.05. The lack of fit value 0.013, 0.080 and 0.014 corresponding to *C. pyralidae* Y1117, *P. kluyveri* Y1125 and *P. kluyveri* Y1164, respectively, implied that the models developed were significant (Appendix A). From the ANOVA analyses, the significance of the models was observed by the values of the lack of fit obtained, which confirmed a good predictability of the models. It was observed that the predicted regression coefficient of 0.9721, 0.8385 and 0.9927 was in reasonable agreement with the adjusted regression coefficient of 0.9897, 0.9787 and 0.9972 for *C. pyralidae* Y1117, *P. kluyveri* Y1125 and *P. kluyveri* Y1164, respectively, which further confirmed a good predictability of the models to observed experimental data.

Appendix A clearly illustrates the comparison of the experimental and the predicted response values of each model developed for the production of potential biopreservation compounds by each yeast. In addition, studentized and normal percentage probability data were also generated to confirm any response transformation. Overall, normality challenges were not encountered. The predicted response for production of biopreservation compounds was carried out using numerous models (Equations (2)–(4)). It is clearly visible from the studentized residuals and actual versus predicted plots that the predicted and the measured (actual) VZI values were comparable. This can further be explained by the alignments of the points close to the slope of the graphs corresponding to each yeast (Appendix A).
(2)$$\begin{array}{ll} Y_{1}=0.69+0.17A & -0.18B+0.018C+0.11D-0.064AB+0.12AC-0.014AD-0.32BC\\ & +0.48BD+0.21CD-0.26A^{2}+0.15B^{2}+0.15C^{2}-0.11D^{2} \end{array}$$
(3)$$\begin{array}{ll} Y_{2}=0.43+0.083A & -0.15B+0.018C+0.065D-0.13AB-0.11AC+0.081AD-0.076BC\\ & -0.044BD+0.073CD-0.11A^{2}+0.039B^{2}+0.0385C^{2}-0.16D^{2} \end{array}$$
(4)$$\begin{array}{ll} Y_{3}=0.34+0.10A & -0.25B-0.031C+0.062D-0.10AB-0.047AC+0.041AD+0.037BC\\ & -0.092BD-0.015CD-0.21A^{2}+0.20B^{2}+0.020C^{2}-0.00495D^{2} \end{array}$$

In the current study, the majority of independent variables were significant terms in the models. The *C. pyralidae* model showed that temperature has less of an effect on biopreservation compound production. The linear effect of time and sugar concentration (including the quadratic effect of sugar) was also insignificant. These observations showed the ability of *C. pyralidae* to produce biopreservation compounds regardless of changes in temperature and sugar concentration. For *P. kluyveri* Y1125, most model terms were significant, except for the linear effect of pH and sugar concentration. However, only the quadratic effect of sugar was insignificant for *P. kluyveri* Y1164 (Appendix A).

Based on the criteria and the boundaries selected for achieving maximum production of biopreservation compounds, a 24 h fermentation period, a temperature of 25 °C, a total sugar concentration of 150 g L^−1^ and a pH of 3 were identified as the optimum conditions for production of potential biopreservation compounds for *C. pyralidae* Y1117 (Appendix A). The optimum production conditions for *P. kluyveri* strains were 28 h of fermentation, pH 3, temperature of 25 °C and sugar concentration of 150 g L^−1^ (Appendix A). These optimization findings were confirmed by the results of the growth inhibition assay (Figure 5). Growth inhibition activity against *D. bruxellensis* before (Figure 5a) and after (Figure 5b) the optimization study was compared and larger inhibition zones were observed after optimization.

### 3.4. Identification and Quantification of VOCs

Many VOCs were detected, but only eight were at a quantification limit. The compounds quantified were isoamyl acetate, butyric acid, isoamyl alcohol, 2-phenyl ethylacetate, hexanoic acid, 2-phenyl ethanol, octanoic acid and decanoic acid. Some of these compounds have been found to have growth inhibition effect on selected fungal pathogens [28,29]. VOCs with inhibition activity, isoamyl acetate, 2-phenyl ethylacetate, isoamyl alcohol and 2-phenyl ethanol [28] were considered to be responsible for growth inhibition activity in this study. The yeasts used were able to produce these compounds at different concentrations (Table 3). Only *C. pyralidae* produced isoamyl acetate below the detection limit. The findings of the current work were in agreement with the current literature reviewed. The exploration of GPE as fermentation medium for the production of biopreservation compounds showed that this growth medium actually allowed a production of isoamyl acetate, butyric acid, isoamyl alcohol, 2-phenyl ethylacetate, hexanoic acid, 2-phenyl ethanol, octanoic acid and decanoic acid at a higher concentration than that reported by Masoud et al. [28] when synthetic refined media was used. “What other suitable, high yield and cost effective substrates can be used for production of biopreservation compounds?” is one of the questions raised in the current literature reviewed that can be answered through the findings of the research reported herein as diluted GPE was found to yield higher quantities of biopreservation compounds. The result obtained in this work further extends the possible use of wild yeasts as a source of biopreservation compounds while using agro-waste (grape pomace extracts) as fermentation medium.

## 4. Conclusions

The use of cheap, readily available agricultural waste (i.e., grape pomace extracts) for the production of potential biopreservation compounds by yeasts presents a cost-effective and realistic alternative to synthetic chemical preservatives. Yeasts have proven to be a potential source of biopreservation compounds with inhibition properties against various spoilage organisms. *C. pyralidae* Y1117 showed growth inhibition activity against *Z. baillii*, *D. bruxellensis* and *D. anomala*, while the two *P. kluyveri* strains only showed inhibition activity against *D. bruxellensis* and *D. anomala.* The production process was optimized for *C. pyralidae* Y1117, *P. kluyveri* Y1125 and *P. kluyveri* Y1164 and the biopreservation compounds produced identified. The models developed for the production of biopreservation compounds under the optimum conditions for each yeast were shown to be appropriate and statistically sound to mathematically explain the production process for potential biopreservation compounds, using GPE as fermentation medium.

## Figures and Tables

**Figure 1 foods-08-00051-f001:**
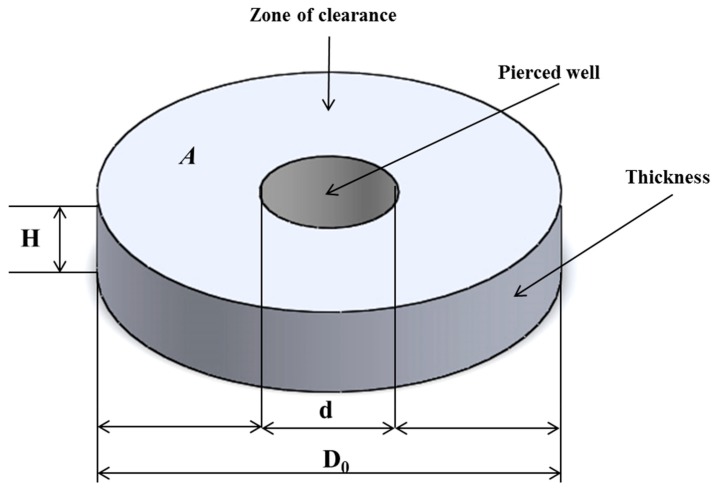
Outline of the concept developed to calculate the volumetric zone of inhibition (VZI) (adapted from Mewa-Ngongang et al., 2017 [23]).

**Figure 2 foods-08-00051-f002:**
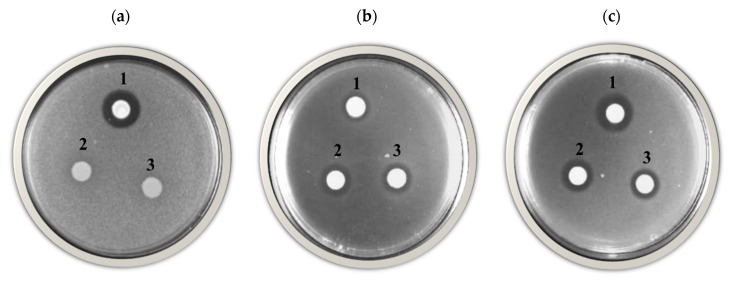
Growth inhibition activity of *Candida pyralidae* Y1117 (1), *Pichia kluyveri* Y1125 (2) and *P. kluyveri* Y1164 (3) against *Zygosaccharomyces baillii* (**a**), *Dekkera bruxellensis* (**b**) and *Dekkera anomala* (**c**) on grape pomace extracts agar medium.

**Figure 3 foods-08-00051-f003:**
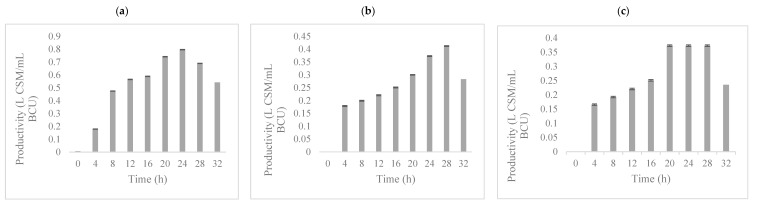
Biopreservation compound production in a grape pomace extracts medium by *Candida pyralidae* Y1117 (**a**), *Pichia kluyveri* Y1125 (**b**) and *P. kluyveri* Y1164 (**c**) in a single stage bioreactor at the total sugar concentration of 150 g L^−1^.

**Figure 4 foods-08-00051-f004:**
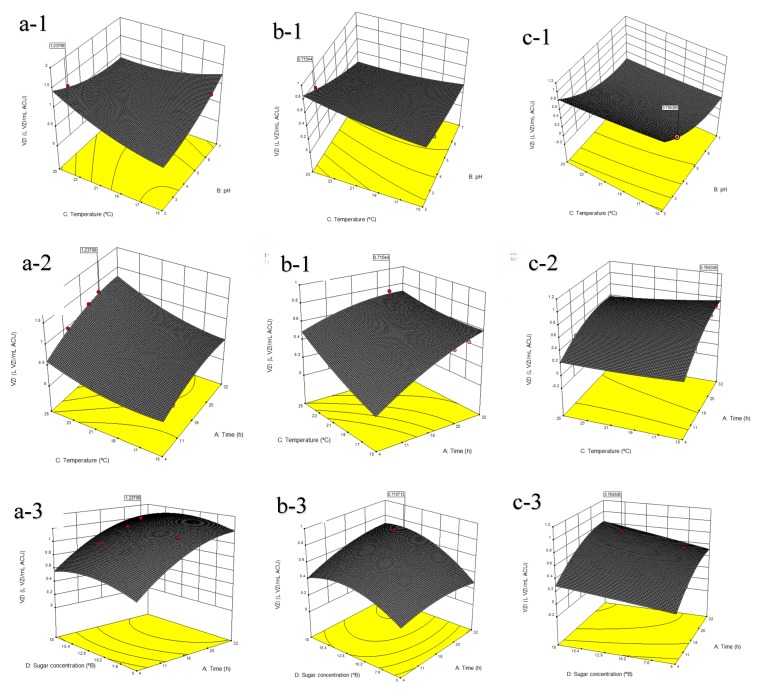
The combined effect of fermentation time and pH (1), time and incubation temperature (2), time and sugar concentration (3) on production of potential biopreservation compounds by *Candida pyralidae* Y1117 (**a**), *Pichia kluyveri* Y1125 (**b**) *and P. kluyveri* Y1164 (**c**). An example: the description, (a-1), represents a = *C. pyralidae* Y1117 and (1) = fermentation time and pH effects.

**Figure 5 foods-08-00051-f005:**
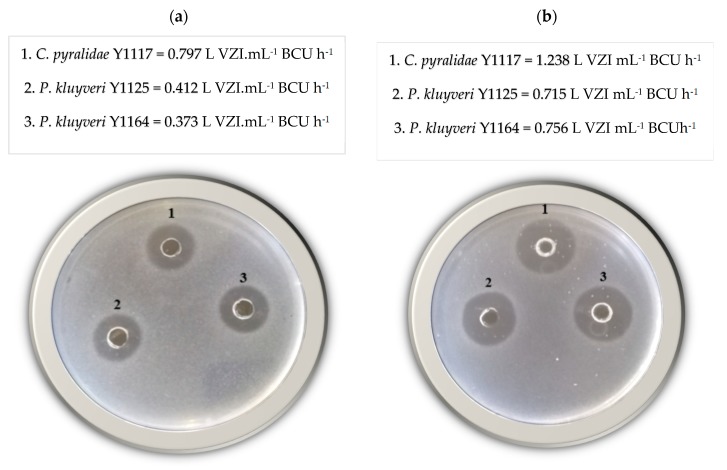
Growth inhibition assay plates showing the inhibition activity against *D. bruxellensis* before (**a**) and after (**b**) optimization of biopreservation compounds production by *Candida pyralidae* Y1117 (1), *Pichia kluyveri* Y1125 (2) and *P. kluyveri* Y1164 (3).

**Table 1 foods-08-00051-t001:** Biopreservation compounds producers and beverage spoilage yeasts selected for the current study.

Biopreservation Compounds Producer Strains	Beverage Spoilage Strains
*Candida pyralidae* Y1117	*Dekkera anomala*
*Pichia kluyveri* Y1125	*Dekkera bruxellensis*
*Pichia kluyveri* Y1164	*Zygosaccharomyces bailii*

**Table 2 foods-08-00051-t002:** Kinetics of biopreservation compound production by *Candida pyralidae* Y1117, *Pichia kluyveri* Y1125 and *P. kluyveri* Y1164 using grape pomace extracts broth (total sugar concentration of 150 g L^−1^) as fermentation medium.

Fermentation Parameters	Model	Antimicrobial Compound Producing Yeasts		
		*C. pyralidae* Y1117	*P. kluyveri* Y1125	*P. kluyveri* Y1164
Substrate (total sugar) utilization rate (g L^−1^ h^−1^)	$r_{s}=\frac{dS}{dt}$	0.333	1.912	1.947
Biomass formation rate (×10^7^ cells mL^−1^ h^−1^)	$r_{x}=\frac{dX}{dt}$	4.542	5.208	3.917
Biomass yield (×10^8^ cells g^−1^)	$Y_{X/S}=\frac{dX}{dS}$	1.365	0.272	0.201
Specific growth rate (h^−1^)	$µ=\frac{ln\left(X_{f}/X_{0}\right)}{t}$	0.196	0.202	0.190
Biopreservation compound formation rate (×10^3^ L VZI mL^−1^ BCU h^−1^)	$r_{p}=\frac{dP}{dt}$	33.209	15.547	15.547
Biopreservation compound formation based on cell concentration (×10^−12^ L VZI cells^−1^)	$Y_{P/X}=\frac{dP}{dX}$	73.121	29.850	39.694
Biopreservation compound formation based on substrate (total sugar) utilization (×10^−3^ L VZI g^−1^)	$Y_{P/S}=\frac{dP}{dS}$	99.840	8.130	7.985
Total sugar utilization rate (g L^−1^ h^−1^) proportional to cellular growth and formation of biopreservation compounds	dSdt=12(dXYXSdt+dPYPSdt)	0.333	1.912	1.947

**Table 3 foods-08-00051-t003:** List of quantified volatile organic compounds (VOCs) identified to be produced by *Candida pyralidae* Y1117, *Pichia kluyveri* Y1125 and *P. kluyveri* Y1164 when grape pomace extract is used as fermentation medium.

	VOCs and Concentrations (mg/L)		
Compound	*C. pyralidae* Y1117	*P. kluyveri* Y1125	*P. kluyveri* Y1164
Isoamyl acetate	not detected	16.51	17.73
Isoamyl alcohol	1.73	1.74	1.89
Butyric acid	1.24	1.25	1.25
2-Phenyl ethylacetate	1.47	1.97	1.99
Hexanoic acid	0.93	0.93	0.93
2-Phenyl ethanol	1.61	1.66	1.68
Octanoic acid	1.32	1.32	1.32
Decanoic acid	1.44	1.44	1.44

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
