# Peer review of "Grape Pomace Extracts as Fermentation Medium for the Production of Potential Biopreservation Compounds"

_foods, 2019, doi:10.3390/foods8020051_

Round 1
Reviewer 1 Report
General comment: The paper of Mewa-Ngongang et al. describe the validation of a relative cheap culture medium for production of biopreservation compounds by yeasts. The underlying algorithms and conclusions thereof appear scientific valid, although hard to understand if not being expertized in fermentation optimization. However some critical points arise from a microbiological point of view. The method for determining the inhibition are not adequate described in the method section as well as the definition of the inhibition activity. In principle it can be assumed that the diameter of inhibition zone was measured which was afterwards converted in a very bulky dimension, which the reviewer cannot follow without some explanations. Furthermore it should be stated that inhibition was exclusively tested with sensitive yeasts, which are of course easier to handle than fungal spoilers. Far more it should be discussed that VOC are only one of several possible reasons for yeast antagonistic activities among others like enzymes or peptide killer toxins (see e.g. Fermentation 2018, 4, 31; doi:10.3390/fermentation4020031). Indeed several literature studies cited by the authors confer to killer factors other than VOC (4, 10, 24). The responsible factors have not been elucidated in the study, although production of various VOCs was demonstrated. This may not be in the focus of the study but should at least be discussed.
Page | Line | Comment |
1 | 22 | The definition of inhibition is hard to understand: what does mean a zone of inhibition (VZI) given as L (liter?) contaminated? (you mean inoculated?) solidified medium (CSM) per ml. The definition should be clearly explained in chapter 2.3 |
3 | 96 | °C |
3 | 102 | The medium contained 0.02 % agar? This would never give a solidified agar medium! |
3 | 104 | What does is mean: A 5 mm diameter wells in GPE plates (what is it?) were seeded with x cells of.. ? Were the cells seeded in the wells, surely not. How were the wells formed in 0.02% liquid agar media? Please add some sentences to clear this passage. |
8 | Table 1 | The dimensions of inhibition are hard to understand: 10^3 L VZI ml^-1 BCH h^-1? or 10^-12L VZI cell^-1? Please explain in chapter 2.3 |
22 | ref. 12 | The reference seem to be uncomplete |
22 | ref. 22 | Food Sci. Technol. Int. |
Author Response
Response to Reviewer 1 Comments
[Foods] Manuscript ID: foods-388591
General comment: The paper of Mewa-Ngongang et al. describe the validation of a relative cheap culture medium for production of biopreservation compounds by yeasts. The underlying algorithms and conclusions thereof appear scientific valid, although hard to understand if not being expertized in fermentation optimization. However, some critical points arise from a microbiological point of view. The method for determining the inhibition are not adequate described in the method section as well as the definition of the inhibition activity. In principle it can be assumed that the diameter of inhibition zone was measured which was afterwards converted in a very bulky dimension, which the reviewer cannot follow without some explanations. Furthermore it should be stated that inhibition was exclusively tested with sensitive yeasts, which are of course easier to handle than fungal spoilers. Far more it should be discussed that VOC are only one of several possible reasons for yeast antagonistic activities among others like enzymes or peptide killer toxins (see e.g. Fermentation 2018, 4, 31; doi:10.3390/fermentation4020031). Indeed several literature studies cited by the authors confer to killer factors other than VOC (4, 10, 24). The responsible factors have not been elucidated in the study, although production of various VOCs was demonstrated. This may not be in the focus of the study but should at least be discussed.
General comments:
Point 1: The method for determining the inhibition are not adequate described in the method section as well as the definition of the inhibition activity
Response: A new section (2.3) is now included, which describes the inhibition activity as well as the concept of volumetric zone of inhibition (VZI) and the related calculations (Please refer to line 96)
Point 2: In principle it can be assumed that the diameter of inhibition zone was measured which was afterwards converted in a very bulky dimension, which the reviewer cannot follow without some explanations
Response: A new section (2.3) is now included, which describes how the inhibition activity as well as the concept of volumetric zone of inhibition (VZI) and the related calculations (Please refer to line 96)
Point 3: It should be stated that inhibition was exclusively tested with sensitive yeasts, which are of course easier to handle than fungal spoilers. Far more it should be discussed that VOC are only one of several possible reasons for yeast antagonistic activities among others like enzymes or peptide killer toxins (see e.g. Fermentation 2018, 4, 31; doi:10.3390/ fermentation 4020031)
Response: A more detailed discussion has been provided (Please refer to line 60-63)
Point 4: Indeed several literature studies cited by the authors confer to killer factors other than VOC (4, 10, 24). The responsible factors have not been elucidated in the study, although production of various VOCs was demonstrated. This may not be in the focus of the study but should at least be discussed.
Response: This comment has been addressed (Please refer to line 200-204)
Specific Comments:
Point 1: Abstract, Line 22: The definition of inhibition is hard to understand: what does mean a zone of inhibition (VZI) given as L (litre?) contaminated? (You mean inoculated.) Solidified medium (CSM) per ml. The definition should be clearly explained in chapter 2.3
Response: The definition of inhibition activity, VZI has been explained in section 2.3 (Please refer to line 96)
Point 2: Material and Methods, Line 96: °C
Response: The correct symbol has been used (Please refer to line 127)
Point 3: Material and Methods, Line 102: The medium contained 0.02 % agar? This would never give a solidified agar medium!
Response: The correct medium concentration (2%) is now used (Please refer to line 133)
Point 4: Material and Methods, Line 104: What does is mean: A 5 mm diameter wells in GPE plates (what is it?) were seeded with x cells of.. ? Were the cells seeded in the wells, surely not. How were the wells formed in 0.02% liquid agar media? Please add some sentences to clear this passage
Response: A sentence clarifying this passage has been provided (Please refer to line 135-139)
Point 5: Results and Discussions, Table 1: The dimensions of inhibition are hard to understand: 10^3 L VZI ml^-1 BCH h^-1? or 10^-12L VZI cell^-1? Please explain in chapter 2.3
Response: The concept behind the VZI has been explained in section 2.3 (Please refer to line 96)
Point 6: Reference 12: The reference seem to be incomplete
Response: Reference 12 has been corrected (Please refer to line 419)
Point 7: Reference 22: Food Sci. Technol. Int.
Response: Reference 22 has been corrected (Please refer to line 444)

Reviewer 2 Report
In this paper, the authors describe the production of inhibitory metabolites by three yeast isolates cultured on grape pomace extracts against three other yeast spoilage isolates using a well diffusion assay. A great deal of detail regarding the variability in production of the metabolites, quantified generally in inhibitory units, is included and the use of SRM to describe production is valuable. Additional details regarding some of the methodology should be provided, specific examples highlighted below.
Additionally, the framing of the context/utility of the findings may require revision. The authors generally evaluated production of biopreservatives in terms of growth inhibition, they do not directly quantify the production of specific metabolites as a function of culture conditions. This limits their ability to draw strong conclusions about the application of this industrial fermentation system given the lack of specific information about what it is they’re biosynthesizing, at what rates, etc. For example, the authors state that the purpose of culturing yeast on grape pomace extracts is a cost-effective way to utilize an agricultural waste stream to produce valuable biopreservative compounds to prevent food spoilage. This could theoretically be true, depending on the identity/nature of the bioactive compound; however, the primary bioactive compounds identified by the authors are acetates and alcohols. It is unclear that the production of these metabolites is specifically limited by a cost-effective growth substrate (considering the production of these compounds in other food fermentations) or that it would be applicable in food preservation. For comparison, the other studies referenced by the authors (23-27) describe biosynthesis and extraction of proteinaceous antimicrobial compounds, like bacteriocins, or in situ production of VOCs like the authors identify during microbial growth, a notably different application model distinct from the use of a waste stream as a growth medium. To improve the manuscript, the authors could consider introducing the identification of the relevant metabolites earlier and refining their description of the purpose of determining production rates for these metabolites.
L33: Re-structure sentence so more clear
L68: Provide table of isolates, source, and if spoilage or producer strains
L110: The methodological detail of section 2.4 is insufficient. Experimental conditions and parameters need to be described for each evaluation (substrate utilization, biomass formation, specific growth, biopreservation compound formation, and biomass yield) to the same degree as the methods in the above sections are described. I appreciate the table of equations provided in S1, but details on how values for the variables were determined is needed.
L159: Name the reference instead of starting the sentence with the reference number
L172: This conclusion is premature, it would depend on several factors about the compound(s) responsible for growth inhibition here and their commercial value, extractability, etc. before making this conclusive recommendation
Fig 2: Remove organism name from title in figure and move to legend. Take out colored background, both as a stylistic preference and to enhance readability for colorblind audiences and black-scale printing.
Table 1 – is info in the “Model” column redundant with supplementary table S1?
L237: Significance of the models? Lack of significance of the fit?
L238: Why are the “pred” and “adj” values provided instead of a statement of what the shorthand model outputs actually are?
Author Response
Response to Reviewer 2 Comments
[Foods] Manuscript ID: foods-388591
In this paper, the authors describe the production of inhibitory metabolites by three yeast isolates cultured on grape pomace extracts against three other yeast spoilage isolates using a well diffusion assay. A great deal of detail regarding the variability in production of the metabolites, quantified generally in inhibitory units, is included and the use of SRM to describe production is valuable. Additional details regarding some of the methodology should be provided, specific examples highlighted below. Additionally, the framing of the context/utility of the findings may require revision. The authors generally evaluated production of biopreservatives in terms of growth inhibition, they do not directly quantify the production of specific metabolites as a function of culture conditions. This limits their ability to draw strong conclusions about the application of this industrial fermentation system given the lack of specific information about what it is they’re biosynthesizing, at what rates, etc. For example, the authors state that the purpose of culturing yeast on grape pomace extracts is a cost-effective way to utilize an agricultural waste stream to produce valuable biopreservative compounds to prevent food spoilage. This could theoretically be true, depending on the identity/nature of the bioactive compound; however, the primary bioactive compounds identified by the authors are acetates and alcohols. It is unclear that the production of these metabolites is specifically limited by a cost-effective growth substrate (considering the production of these compounds in other food fermentations) or that it would be applicable in food preservation. For comparison, the other studies referenced by the authors (23-27) describe biosynthesis and extraction of proteinaceous antimicrobial compounds, like bacteriocins, or in situ production of VOCs like the authors identify during microbial growth, a notably different application model distinct from the use of a waste stream as a growth medium. To improve the manuscript, the authors could consider introducing the identification of the relevant metabolites earlier and refining their description of the purpose of determining production rates for these metabolites.
General comments:
Point 1: To improve the manuscript, the authors could consider introducing the identification of the relevant metabolites earlier and refining their description of the purpose of determining production rates for these metabolites.
Response: We do appreciate the reviewer’s comment. However, the initial order was important since it was crucial to firstly identify the suitable medium for the selected yeasts, the production profiles as well as the growth inhibition activity before the characterisation of potential compounds responsible.
Specific Comments:
Point 2: Abstract, Line 33: Re-structure sentence so more clear
Response: The sentence has been re-structured (Please refer to line 33)
Point 3: Material and Methods Line 68: Provide table of isolates, source, and if spoilage or producer strains
Response: A table with the required details has been provided (Please refer to line 85)
Point 4: Material and Methods, Line 110: The methodological detail of section 2.4 is insufficient. Experimental conditions and parameters need to be described for each evaluation (substrate utilization, biomass formation, specific growth, biopreservation compound formation, and biomass yield) to the same degree as the methods in the above sections are described. I appreciate the table of equations provided in S1, but details on how values for the variables were determined is needed
Response: The methodological details of section 2.4 (now section 2.5) has been provided (Please refer to line 141-152)
Point 5: Results and Discussions, Line 159: Name the reference instead of starting the sentence with the reference number
Response: As suggested, the reference has been named (Please refer to line 195)
Point 6: Results and Discussions, Line 172: This conclusion is premature, it would depend on several factors about the compound(s) responsible for growth inhibition here and their commercial value, extractability, etc. before making this conclusive recommendation
Response: The conclusion was corrected (Please refer to line 211-213)
Point 7: Results and Discussions, Figure 2: Remove organism name from title in figure and move to legend. Take out colored background, both as a stylistic preference and to enhance readability for colorblind audiences and black-scale printing.
Response: The figure has been adjusted accordingly (Please refer to line 226)
Point 8: Results and Discussions, Table 1: is info in the “Model” column redundant with supplementary table S1?
Response: Supplementary table S1 describes the fermentation parameters applicable to each model listed. However, table 1 (now table 2) presents the actual values obtained when the experimental data were fitted in each of the model described in supplementary table S1. It was therefore important to have the models in both tables.
Point 9: Results and Discussions, Line 237: Significance of the models? Lack of significance of the fit?
Response: The sentence has been rephrased, and more clarity has been given (Please refer to line 276-278)
Point 10: Results and Discussions, Line 238: Why are the “pred” and “adj” values provided instead of a statement of what the shorthand model outputs actually are?
Response: The predicted and adjusted R2 values are provided in order to show the significance and predictability of the models developed (Please refer to line 276-281)
